**Data Availability Statement:** All quantitative datasets used in this are available from the Dryad database (DOI https://doi.org/10.5061/dryad.bcc2fqzbj). The qualitative data (transcripts of

# Quality of care in a differentiated HIV service delivery intervention in Tanzania: A mixed-methods study

Nwanneka Ebelechukwu Okere[1]*, Judith Meta[1], Werner Maokola[2], Giulia Martelli[3], Eric van Praag[1], Denise Naniche[4], Gabriela B. Gomez[5], Anton Pozniak[6], Tobias Rinke de Wit[1], Josien de Klerk[1], Sabine Hermans[1]

1 Amsterdam Institute for Global Health and Development (AIGHD), Department of Global Health, Amsterdam UMC, University of Amsterdam, Amsterdam, Netherlands, 2 Department of Strategic Information, National AIDS Control Programme, Dodoma, Tanzania, 3 Doctors with Africa – CUAMM, Padova, Italy, 4 ISGlobal -Barcelona Institute for Global Health, Hospital Clinic, University of Barcelona, Barcelona, Spain, 5 Department of Global Health and Development London School of Health and Tropical Medicine London, London, United Kingdom, 6 Chelsea and Westminster Hospital NHS Foundation Trust, and LSHTM London, London, United Kingdom

☯ These authors contributed equally to this work.

* n.okere@aighd.org

## Abstract

### Background

Differentiated service delivery (DSD) offers benefits to people living with HIV (improved access, peer support), and the health system (clinic decongestion, efficient service delivery). ART clubs, 15–30 clients who usually meet within the community, are one of the most common DSD options. However, evidence about the quality of care (QoC) delivered in ART clubs is still limited.

### Materials and methods

We conducted a concurrent triangulation mixed-methods study as part of the Test & Treat project in northwest Tanzania. We surveyed QoC among stable clients and health care workers (HCW) comparing between clinics and clubs. Using a Donabedian framework we structured the analysis into three levels of assessment: structure (staff, equipment, supplies, venue), processes (time-spent, screenings, information, HCW-attitude), and outcomes (viral load, CD4 count, retention, self-worth).

### Results

We surveyed 629 clients (40% in club) and conducted eight focus group discussions, while 24 HCW (25% in club) were surveyed and 22 individual interviews were conducted. Quantitative results revealed that in terms of structure, clubs fared better than clinics except for perceived adequacy of service delivery venue (94.4% vs 50.0%, p = 0.013). For processes, time spent receiving care was significantly more in clinics than clubs (119.9 vs 49.9 minutes). Regarding outcomes, retention was higher in the clubs (97.6% vs 100%), while the

interviews and focus group discussions) contained potentially identifying participants' information and were not included in the dryad dataset. The qualitative data can be accessed upon reasonable request from the Shinyanga T & T project scientific committee through secretariat@aighd.org.

**Funding:** The Shinyanga and Simiyu Test & Treat program in Tanzania is funded by Gilead Sciences (USA) and the Diocese of Shinyanga through the Good Samaritan Foundation (Vatican). ONE was funded by the Erasmus Mundus Joint Doctorate Trans Global Health Programme EMJD-TGH (Framework Partnership Agreement 2013-0039, Specific Grant Agreement 2014-0681) http://www.transglobalhealth.org/ and the Amsterdam Institute for Global Health and Development (AIGHD). The funders had no role in study design, data collection, and analysis, decision to publish, or preparation of the manuscript.

**Competing interests:** The Shinyanga and Simiyu Test & Treat program in Tanzania is funded by Gilead Sciences (USA). This does not alter our adherence to PLOS ONE policies on sharing data and materials. [GBG is currently employed by Sanofi Pasteur. Sanofi Pasteur was not involved in any way and did not provide funding for this study]. All other authors declare that they have no competing interests.

proportion of clients with recent viral load <50 copies/ml was higher in clinics (100% vs 94.4%). Qualitative results indicated that quality care was perceived similarly among clients in clinics and clubs but for different reasons. Clinics were generally perceived as places with expertise and clubs as efficient places with peer support and empathy. In describing QoC, HCW emphasized structure-related attributes while clients focused on processes. Outcomes-related themes such as improved client health status, self-worth, and confidentiality were similarly perceived across clients and HCW.

## Conclusion

We found better structure and process of care in clubs than clinics with comparable outcomes. While QoC was perceived similarly in clinics and clubs, its meaning was understood differently between clients. DSD catered to the individual needs of clients, either technical care in the clinic or proximate and social care in the club. Our findings highlight that both clinic and DSD care are required as many elements of QoC were individually perceived.

## Introduction

Quality of care is at the heart of the differentiated care strategy currently endorsed by WHO for HIV programs. The core principles underlying the approach include client-centeredness and health system efficiency [1], both of which constitute important dimensions of quality [2]. The design of many differentiated service delivery (DSD) interventions reflects these values by prioritizing the needs of clients while considering the health system characteristics. DSD interventions are conducted both within health facilities and the community and rely on formally trained health care workers (HCW), peers, and community health workers (CHW). The involvement of peers and CHW with varying degrees of formal training as an essential part of DSD warrants further investigation to ensure that quality is not compromised.

Community health workers (CHW) have been involved in various HIV interventions before DSD roll-out [3–6]. Their role in DSD varies depending on whether they are supporting or coordinating the specific intervention. In a supportive role, they assist other HCW to provide adherence counseling, distribute pre-packaged antiretrovirals (ARVs), client tracking, documentation, and home visits. As coordinators, they are responsible for facilitating antiretroviral therapy (ART) "clubs" (i.e., small groups of 15 to 30 stable clients who meet at the clinic or community), screening and identifying symptoms of common opportunistic infections e.g., tuberculosis (TB) for upward referral, following up clients who miss appointments, collecting and distributing ARVs to clients. Though good outcomes have been reported with CHW playing these expanded roles, evidence is sparse on the quality of care (QoC) provided in these CHW-led DSD interventions [7–9].

In Tanzania, CHW are involved in health promotional, educational, and rehabilitative interventions but their role beyond these activities, in particular in providing basic curative services, is yet to be formalized. Several studies show promise for expanding CHW roles, but more evidence is warranted [6, 10]. Since DSD limits the frequency of clinical encounters for clients, it becomes pertinent to assure the QoC provided by these lay providers in order not to compromise client outcomes.

Generally, quality underscores the goal of many health systems. However, the complex, subjective and multi-dimensional nature of quality care makes it a difficult concept to define and

therefore measure [11]. The Donabedian framework is arguably the most widely used to assess QoC [11, 12]. It promotes a three-pronged approach to assessing QoC encompassing **structure** (characteristics of the care setting and resources available e.g., staff, equipment, supplies, venue), **process** (activities conducted in care provision e.g. time-spent, health screenings, HCW attitude, information), and **outcome** (the effects of care on care recipients e.g. viral load, CD4 count, improved self-worth/confidence and health status) [12, 13].

CHW-led DSD clubs for clients stable on ART (see definition below) have been piloted at the Test and Treat (T&T) project sites in the Shinyanga region, north-western Tanzania, since July 2018. Details of the implementation and research projects have been published elsewhere [14]. This study sought to assess QoC in terms of the sub-themes of structure, process, and outcome of care as outlined by the Donabedian framework. Primarily, we aimed to describe the structures supporting services delivery and the processes of care, to assess some objective client-related outcomes as well as gain the perspectives of clients and HCW, comparing between the DSD clubs and standard clinic care. Our study contributes evidence to the quality and effectiveness of these CHW-led interventions with implications for the scale-up of the DSD strategy.

## Materials and methods

### Study design and outcomes

A concurrent triangulation mixed-method study design was employed to facilitate the simultaneous assessment of the quality of HIV care employing client-related outcomes and exploring perspectives of both clients and HCW [15–17]. The quantitative part entailed cross-sectional surveys of stable ART clients and HCW in the clinics and clubs. The qualitative part entailed focus group discussions among clients and individual interviews among HCW.

We organized our findings according to the three domains of the Donabedian framework i.e., Structure, Process, and Outcome. Within each domain, we first reported clients' experience of care in the clinic and club, and then the HCW's experience in a similar fashion. In each section, we presented the quantitative, followed by the qualitative findings, as the latter triangulated and provided a deepened understanding of the former. Finally, we summarized the main findings in a joint display table [18].

### Study sites

The study was conducted at two HIV care and treatment centers (CTC) owned by the Catholic diocese in the Shinyanga region, Tanzania. Bugisi CTC serves a large widely dispersed population in rural Shinyanga district while Ngokolo CTC serves a peri-urban population in Shinyanga municipality. Both health centers coordinate ART clubs in proximal communities within distances ranging from 3 to 35km. As of June 2019, the time of study commencement and about a year since the commencement of the clubs, there were 46 clubs in total, 25 of which were considered eligible for our study as they had existed for at least 6 months and had a club meeting scheduled within the data collection period (see Table 1).

### Study sampling procedure

Stable ART clients were sampled from both clinics and eligible clubs and all HCW providing care at the clinics and clubs. Stable ART clients were defined per the Tanzanian HIV care and treatment guideline as those above five years of age, having received ART for at least six months with >95% adherence and no adverse drug reaction or current illnesses [19]. For our study, we included only adults≥18 years old. Sample size calculation for the quantitative part

**Table 1. Numbers of interviewed participants per location and data collection method.**

| | Clinic | Club | Total |
|---|---|---|---|
| **Location** | | | |
| • Bugisi (Rural) | 1 | 16 | |
| • Ngokolo (Peri-urban) | 1 | 9 | |
| **Quantitative** | | | |
| Survey—Clients | 378 | 251 | 629 |
| Survey—Healthcare workers (HCW) | 18 | 6 | 24 |
| **Qualitative** | | | |
| Focus Group Discussion (FGD) participants | **23** | **18** | 41 |
| • Female | 12 | 9 | |
| • Male | 11 | 9 | |
| Number of FGD with clients | 4 | 4 | 8 |
| Individual Interviews with HCW | 16 | 6 | 22 |

of the study estimated a total of 334 participants (167 per group), assuming an effect difference of 14% between clinics and clubs using retention in care as a proxy for QoC and an alpha of 0.05 and 80% power [20]. In the absence of an accepted quantitative measurement of QoC, we used retention as a measure for QoC assuming that clients are more likely retained in care when the QoC was acceptable. We defined retention as attendance to the last three clinic appointments/club meetings within the past 9–12 months, given that clubs held quarterly, and the oldest clubs were just about one year old and it was too soon to measure one-year retention across all clubs. Clients were recruited as they attended clinics and clubs for the survey and focus group discussions. At the clinics, a random list of eligible clients scheduled for an appointment was generated on every clinic day. Clients were approached as their number appeared on the random list and those who gave written consent participated. At eligible clubs, all clients were approached during the routine club meetings and those who gave written consent participated. Similarly, all HCW were approached and those who consented were recruited to participate in the survey and individual interviews.

## Description of ART club intervention

ART clubs were commenced at the two CTC study sites in July 2018. Stable ART clients living within the same community are invited to constitute clubs. Details of the club model have been described elsewhere [14]. Briefly, under the supervision of the designated nurse, a CHW from the CTC liaises with the existing home-based care worker (HBC) of the community to coordinate club meetings. Club meetings hold every 3 months in community venues selected by members which could be homes, school classrooms, and community halls. At the meetings, the CHW conducts a health talk followed by adherence counseling, weight taking, TB/other infections screening, and drug distribution. Any member with symptoms requiring further investigation was referred to the CTC and the referral was documented appropriately in the club register and client folder.

## Description of clinic-based care for stable clients

Stable ART clients who received care in the clinics were seen every 2–3 months at the discretion of the clinician. On clinic days, they go through group counseling/health talk, triage, clinical consultation, and lastly drug pick-up.

## Quantitative data collection and analysis

For clients, we adapted an existing instrument, the QoC from the clients' perspective—QUO-TE-HIV [21]. The 27-item QUOTE-HIV instrument covers clients' perspectives on generic and HIV-specific aspects of the quality-of-service delivery. Seventeen items were retained as is, four items were combined to make two, four items were rephrased and four items were added to make it more contextually relevant (see S1 Appendix: QoC questionnaire-English & S2 Appendix: QoC questionnaire II-Swahili). For HCW, appropriate questions exploring the structure of care were developed. The Donabedian framework served as a general guide for developing all study questions. Additional data to assess processes and outcomes of care were extracted from client records for the three most recent visits/club meetings e.g., visit attendance, weight measurement, infections/TB screening, referrals, ARV dispensed, adherence assessment, CD4 count, and viral load test and results. Other process-related factors collected in the survey included respectful service and time spent during service.

Clients and HCW participants were characterized and compared between clinics and clubs using the Mann-Whitney test. In terms of the structure of care, the availability of resources for the provision of services was described i.e., human, physical, and financial resources as well as organization and information management. We assigned a value of 0 and 1 to every negative and positive response respectively and summarized percentage scores between clinic and club in the sub-categories. The items in the adapted QUOTE-HIV were categorized in terms of structure, process, and outcome of care as per the Donabedian framework [11]. Proportions of clients reporting their experience of care as "always", "mostly", "occasionally" or "never" across the adapted QUOTE-HIV items were compared between clinic and club according to the Donabedian framework using the Chi-squared test. Main outcomes of care were compared between clinic and club i.e., proportions with suspected opportunistic infection or TB, most recent CD4 count (cells/mm3), and viral load > 50 copies/ml (defined as <12 months). To reduce the probability of ascribing an association as significant when, in fact, it was not (i.e., Type 1 error) due to the multiple comparisons made, we lowered the significance level appropriately using the Bonferroni correction. Therefore, considering the 30 variables used in our study and alpha of 0.05, only p values $\leq 0.002$ were considered significant. All quantitative data were doubly entered, validated, and managed using EpiData software and analyzed using STATA 16 and MS Excel.

## Qualitative data collection and analysis

We employed three data collection methods for the qualitative part of the study: focus group discussions (FGD) with clients, in-depth interviews (IDI) with HCW using a semi-structured guide, and a structured observation tool of club meetings. The interview/discussion guide queried participants' perceptions of QoC (structure, process, and outcome), the benefits, and suggestions for improving and sustaining the clubs. The observation guide enabled the detailed articulation of activities in clinics and clubs such as the venue, client characteristics, topics discussed, and interactions among clients and facilitators. Among clients, eight FGD were conducted in Swahili by two trained research assistants with one facilitator and the other taking notes. Participants were sampled from among clinic and club clients who consented to participate in the survey. To facilitate communication, FGD were conducted segregated by sex, four with men alone (two among clinic participants and two among club participants) and the other four similarly with women alone. To make the abstract concept of quality understandable, discussions started asking for participants' preference between two different African textile fabrics "Kitenge", one of which was regarded as highly valuable. Discussions around the reason for preferring one fabric over the other made it easier to introduce the topic of QoC.

Interviews were conducted in either Swahili or English by trained research assistants (see Table 1). Thematic analysis was employed to analyze the qualitative data. An iterative process of reading, transcription, and translation of FGD and IDI memos was used to inductively develop a codebook using the NVIVO 12 Plus software version. In the results inductively derived themes were grouped under the broader Donabedian framework of structure, process, and outcome of care to facilitate comparison between quantitative and qualitative results. Differences in perspective between participants in the clinic and clubs were explored.

## Ethical consideration

This study was approved by National Institute for Medical Research, Tanzania (Reference Number NIMR/HQ/R.8c/Vol. 1/674). All participating clients and HCW provided written informed consent for the survey, interview, and audio recording.

## Results

### Characteristics of study participants

Details of the socio-demographic characteristic of participants comparing between clinic and club in the clients and HCW survey are presented in Table 2. All results will be presented in the order of clinic vs club. Of 629 participants among consenting clients in the survey, 251 (40%) were accessing care in clubs. While females constituted 62.9% of all participants, there were significantly more male participants in clinics than in clubs (41 vs 32%). Over 80% of participants were aged between 25 and 54 years, with the mean age being older in the club. Educational level was generally low with only 5% above primary. The majority were either married or widowed, separated and divorced.

Characteristics of 24 HCW surveyed are provided in Table 2. All HCW who provided care in the clubs accepted to participate in the survey. Across all HCW cadre and on average, clinic staff were older (43.2 vs 32.8 years) and had worked longer in HIV service delivery compared to club staff. All HCW participants had basic primary level education with about half having above secondary level. The number of clients attended to by each HCW at the clinics and clubs varied with an average of 53 vs 27, respectively.

Forty-one stable ART clients consented to participate in focus group discussions, among these, 23 (56%) were clinic participants and 49% were male. Conversely, 22 HCW consented to be interviewed individually, among these, 72% were clinic staff and 50% were male.

### Structure of care

**Clients.** *Quantitative*. Clients agreed that HCW have basic HIV knowledge by affirming the ability of HCW to answer any questions they had about HIV in both club and clinic (75% vs 76%—Table 3). Club participants however had more access, when necessary, via phone to their HCW compared to clinic participants. Only 70% of clients each in both clinic and club confirm always receiving their supply of ARVs, while the rest report receiving ARVs most of the time. Similarly, despite the inadequate space reported in some clubs, more club clients affirmed that there was adequate space in which to discuss confidentially with HCW. Clients in clubs conceded that HCW who deliver services to them have a good relationship with each other (70.9 vs 73.3%).

*Qualitative*. Related to the structure of care, clients described quality of care in terms of the following themes.

*The provision of expertise and support*: For clients who attended the clinic, the availability of medicine was considered quality care. This group of clients also described the clinic as a

**Table 2. Characteristics of study participants (clients and healthcare workers).**

| Characteristics | Clients | | | Health care workers (HCW) | | | |
|---|---|---|---|---|---|---|---|
| **a. Sociodemographic and clinical profile of Clients** | | | | **b. Sociodemographic profile of HCW** | | | |
| | **Clinic** | **Club** | **p-value** | | **Clinic** | **Club** | **p value** |
| Location n, % | | | <0.001 | | | | 1.000 |
| • Bugisi | 324, 65.8 | 168, 34.1 | | | 10, 55.6 | 4, 66.7 | |
| • Ngokolo | 54, 39.4 | 83, 60.6 | | | 8, 44.4 | 2, 33.3 | |
| Sex n, % | | | 0.018 | | | | 0.640 |
| • Female | 224, 59.3 | 172, 68.5 | | | 8, 44.4 | 4, 66.7 | |
| • male | 154, 40,7 | 79, 31.5 | | | 10, 55.6 | 2, 33.3 | |
| Age in years Mean (SD) Age-groups n, % | 41.0 (11.2) | 46.0 (11.4) | <0.001 | Age in years Mean (SD) | 43.2 (10.8) | 32.8 (9.2) | 0.048 |
| • <25 | 25, 6.6 | 6, 2.4 | <0.001 | Age-group n, % | 0 | 2, 33.3 | |
| • ≥25–34 | 96, 25.1 | 35, 13.9 | | | | | |
| • ≥35–44 | 137, 36.2 | 91, 36.3 | | | | | |
| • ≥45–54 | 75, 19.8 | 62, 24.7 | | • <25 | 16, 88.9 | 4, 66.7 | |
| • ≥55–65 | 33, 8.7 | 40, 15.9 | | • ≥25–55 | 2, 11.1 | 0 | |
| • >65 | 13, 3.4 | 17, 6.8 | | • >55 | | | |
| Educational level n, % | | | 0.801 | Educational level n, % | | | 0.514 |
| • No education | 97, 25.7 | 60, 23.9 | | • *Primary | 6, 33.3 | 0 | |
| • Primary | 261, 69.1 | 180, 71.7 | | • Secondary | 3, 16.7 | 3, 50.0 | |
| • ≥Secondary | 20, 5.3 | 11, 4.4 | | • Certificate/Diploma/Degree | 9, 50.0 | 3, 50.0 | |
| Marital status n, % | | | 0.321 | HCW cadre n, % | | | 0.410 |
| • Single | 94, 24.9 | 80, 31.9' | | • CHW/HBC/DC | 10, 55.6 | 4, 66.7 | |
| • Married | 144, 38.1 | 78, 31.1 | | • Nurse/NA | 3, 16.7 | 2, 33.3 | |
| • Separated/Divorced/Widowed | 140, 37.0 | 93, 37.1 | | • Laboratory Technician | 1, 5.6 | 0 | |
| | | | | • Pharmacy Technician | 2, 11.1 | 0 | |
| Employment status n, % | | | 0.002 | • Doctor/MO | 2, 11.1 | 0 | |
| • Unemployed | 53, 14.0 | 60, 23.9 | | | | | |
| • Employed | 325, 86.0 | 191, | | | | | |
| Years on ART Median (IQR) Years on ART | 3.4 (2.1–5.8) | 4.2 (2.2–7.3) | 0.001 | ^Years in HIV service | | | 0.007 |
| • >2 years | 90, 24.4 | 47, 19.3 | 0.162 | • Median (IQR) | 6 (3–8.5) | 1 (1–2) | |
| • ≤ 2 years | 279, 75.6 | 197, 80.7 | | | | | |
| Time spent during last 3 visits/meeting | 119.9 (75.0- | 49.9 (33.3- | <0.001 | Patients attended daily | | | 0.026 |
| • Median (IQR) | 180.0) | 76.6) | | • Median (IQR) | 50 (25–80) | 27 (15–30) | |

SD—Standard deviation; IQR—Interquartile range; CHW—Community Health Workers; HBC—Home-based Care worker; DC—Data Clerk; NA—Nursing Assistant.

*HCW with Primary education were HBC and CHW;

^The HCW with longer years in service were mostly Doctors and Nurses.

technical space where laboratory tests for HIV and specialized knowledge on other illnesses were available. In contrast, club participants did not consider the availability of HCW, medicines, and lab tests to define quality care. Amongst both Clinic and Club participants, the provision of adjunct non-medical services such as free breakfast and the prospect of receiving food items and monetary aids had an overwhelmingly positive influence on the perception of quality. Though such services were provided only at the T&T project clinic sites (in some sites to all clients and others only to selected poor clients) and not in the clubs, they were still mentioned as part of quality care by club participants.

**Table 3. Structure of care: Clients' perspective on care experience.**

| | Clinic (N = 378) | Club N = 251 | p value |
|---|---|---|---|
| *Care experience with HCW from clients' perspective—Clients' survey n, %* | | | |
| Ensures I get my ARV supply regularly and conveniently (ARV supply) | 276, 73.0 | 179, 71.3 | 0.640 |
| Can answer any questions I have about HIV (HIV education) | 283, 74.9 | 190, 76.0 | 0.814 |
| Works well with other health workers (Interprofessional relationship) | 268, 70.9 | 184, 73.3 | 0.511 |
| I can talk undisturbed during consultation (Confidential space) | 297, 78.6 | 196, 78.1 | 0.885 |
| Easily accessible by telephone (HCW availability) | 162, 42.9 | 158, 62.9 | <0.001 |
| The meeting space is arranged in such a way that no one can hear when I am talking with her in confidence (Confidential space) | 272, 71,9 | 185, 74.1 | 0.554 |

"Service (in the club) is good but if they will have a chance, they can give us nutritional support like flour, sugar, and cooking oil. Because for other people it is difficult to have all of these needs, it is a great problem, because of economic status although we are different in the ability to working."

*Male club participant*

*Clinic decongestion*: The decongestion of the clinic was mentioned by club participants as the motivation for club preference, but club members also choose the club because it saved travel time and therefore allowed time for other activities and also saved travel costs

"First the congestion, we also get time in the morning to do all the work then come to the club in the evening, you get good services, there are no congestion issues like before (in the clinic)"

*Female club participant*

*The role of venue*: While no specific reference was made to aspects of physical space as an issue in the clinic, it was important for clients to have a venue that was permanent and could assure confidentiality. Club clients mentioned that the venue can be a place in the community but could also be a room in the clinic that does not arouse suspicion when clients visit. The infrastructure of the clinic was seen as safeguarding privacy, where clients were not known to other community members and for clinic clients, this was a reason not to transfer to clubs. We found that more male participants opted to continue clinic-based care than ART clubs (40.7 vs 31.5% see Table 2) because they did not feel assured of confidentiality in the venues used for club meetings within their community.

"I will not join a club because in the street [village] is not good, I'm receiving good service here. In villages, it will be like an advertisement"

—*Male clinic participant*

The same value of privacy pertained to club participants but was solved by selecting a venue in the village where it is 'normal' to see people going in and out, such as the HBC's house which is a regular meeting space for many groups.

"We are taking/picking drugs inside the house [of the HBC], a venue [like this] is good it looks like we are coming to this family for normal issues"

—*Female club participant*

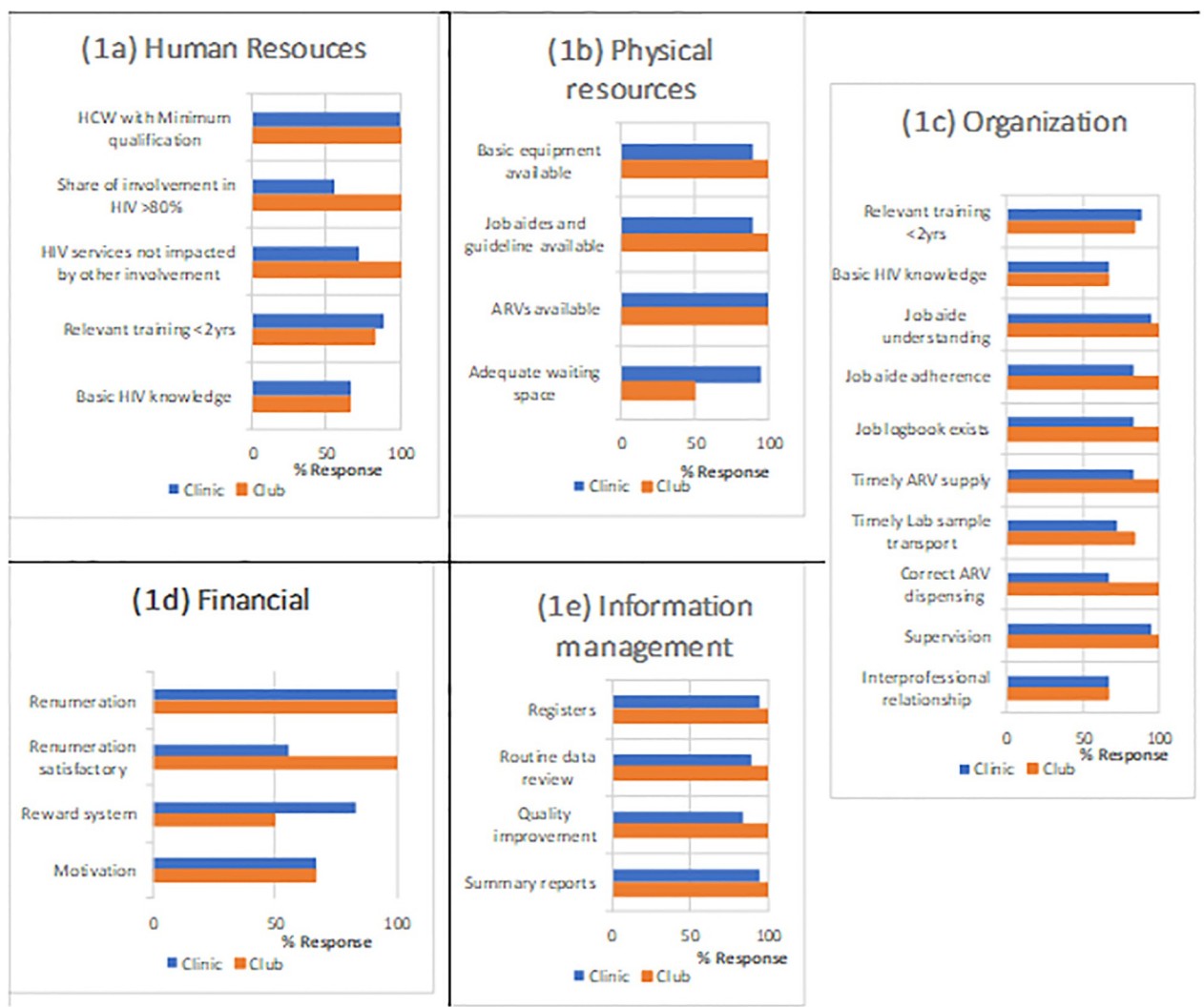

**Fig 1. a-e**: Structure of care: Health Care workers perception.

**Healthcare workers.** *Quantitative.* The possession of minimum job qualification required was comparable between clinic and club staff. Over 80.0% of participants received some form of training in the past 2 years equipping them for their duties, many of which were on-the-job. While all club staff reported maximum involvement in HIV services, 56% of clinic staff reported >80% involvement in HIV services with 72% of those affirming that involvement in other departments did not affect the quality of HIV services they provide. Basic HIV knowledge as well as receiving relevant training within the last two years was similar across clinic and club (see Fig 1a).

All club staff affirmed the availability of basic equipment, materials, job aids, and guidelines required to do their jobs while only 89% among clinic staff affirmed comparably. Adequate ARV availability was reported in clinic and club alike. While sitting and waiting space was reported as adequate in the clinic, in about half of the clubs (94.4 vs 50%), staff reported inadequate space for club meetings (see Fig 1b).

All club staff affirmed understanding, adhering to guidelines, and documenting job done, compared to a slightly lower percentage of clinic staff (all p > 0.05). A similar trend was observed in the timely supply and correct dispensing of ARV, timely transportation of laboratory samples, and periodic supervision for job support. Satisfactory inter-professional relation with a clear understanding of the system in place for conflict resolution was comparable between clinic and club though reported by only two-thirds of HCW (see Fig 1c).

While all HCW reported receiving regular remuneration (see Fig 1d), only about half of clinic staff admitted to being satisfied with their pay. Club staff, who were mostly project staff and likely receiving higher remuneration than their clinic counterpart still gave figures much higher than their current pay as their ideal salary expectations. Though reporting satisfaction with their pay, motivation to continue performing on their job among club staff was similar with clinic staff at only 67%. A higher percentage of clinic staff were aware of a reward system to motivate high-performing staff.

The data management system was largely paper based for both the clinic and club. HCW admitted to documenting client information in registers and making summary reports which are shared usually quarterly with the regional authorities. Routine data review by all HCW to identify areas for quality improvement was also comparable between clinic and club (see Fig 1e.–all p >0.05).

*Qualitative*. Related to structure of care, HCW described quality of care in terms of the following themes:

*Specialized skills*: The clinic had different cadres of health care workers (nurses, clinicians, etc., while the clubs had only community health workers overseen by a club nurse. Quality HIV care was described in terms of having different HCW with specific specialized skills to attend to clients in the clinic, and having HCW tasked to provide tailored care in the club:

> "First, I think I can say a person may come sick and get served concerning what problem he mentioned like for example he says he has a stomach ache, this and this, but then the doctor sees that according to the explanations, then he will have to check this and this, you then go to the laboratory, then later you get the results and according to those results, the doctor writes down the medicines to be taken for that problem and you go to the pharmacist and you get the medicine then that is quality service. . .as such excellent service is the one when you reach and explain yourself is what you go and test and then get the right results, get the right dose, I think that is excellent service"

> —*Male clinic HCW*

*Clinic decongestion, and venue*: HCW in the clinic noted the decongestion of the clinic since the roll-out of clubs as good for facilitating quality care. In the clubs however, HCW mentioned the need for more permanent venues spacious enough to ensure confidentiality.

*Training, collaboration & inter-professional relationship*: Clinic and club staff felt they had enough supervision and support to perform their duties and had good working relations and mutual respect irrespective of cadre.

> "Everyone performs his/her duty as per her/his level, if my level is the nurse or the client I will treat them depending to their levels, I cannot do what is out of my order/level, maybe if I go and give the client medicines that is impossible, the order must be followed, when he reaches and is sick I will look for his file then take him to the doctor and the doctor will do her/his responsibilities so we collaborate and work together"

> *Male clinic HCW*

At the same time, both clinic and club participants felt that continuous training was important. HCW said the quality of their care would improve with increased salary and more off-site training opportunities

"First of all, of course, a satisfactory salary is important. The second one we must have a lot of training seminars yes because for six eight-nine months [we were] without any training"

*Male clinic HCW*

*Real-time documenting*: Documenting the services rendered in appropriate registers or client folders was part of routine service delivery for HCW. Real-time documentation, meant to curtail the cycle of missing or invalid documentation was cited as an important part of providing quality service, as was having a structured and organized filing system. This was however lacking sometimes.

"The filing system, for example currently you can go look for a file but won't get it, it doesn't have any label, they are just there for a new person to get a file it takes a long time"

*Female clinic HCW*

## Process of care

**Clients.** *Quantitative*. From the clients' perspective (see Fig 2a), regarding the practice of care, more clients confirmed the care process in clubs to be both time-saving—clinic vs club (65 vs 79%) while making enough time for personalized caregiving (72 vs 80%).

Most clients in clubs admitted getting referrals when necessary and data extracted from the records also confirm that most clients (93%) of club clients screened eligible received appropriate referrals to the hub. This is excluding women who became pregnant while in the club, who were all referred back to the hub.

Clinic participants see the clinician during a visit while club participants are referred to the clinic.

On average, time spent in the clinic was over double that spent in the club (119.9 vs 49.9 minutes), thus affirming the time-saving manner of service provision in clubs (see Table 4).

Information related activities were perceived comparably by clients e.g., while more clinic clients agreed that their HCW tell them anything they want to know about their health (67.9 vs 64.5%), more club clients stated that their HCW explained benefits, side effects, and prevention strategies (71.2 vs 73.3%—see Fig 2a).

Health monitoring screenings e.g., the proportion of clients who had weight measurements and adherence assessment were comparable in clinic and club. A shortfall was seen in the proportion of clients documented as screened for OI or TB in the club (85.4 vs 78.5% p = 0.024). The proportion of clients who had a recent viral load test done (99.2 vs 97.6% p = 0.081) was similar in clinic and club (Table 4).

More club participants acknowledged having enough individualized attention talking about any issue of personal importance (70.4 vs 79.6%). Interestingly, all clients admit to being served respectfully in the survey—see Table 4. Club clients more readily admit that HCW uses simple terms in explaining things like side effects (see Fig 2a).

*Qualitative*. From both the qualitative interviews, themes related to processes of care commonly used by clients and HCW to describe QoC can be summed into; the practice of care (referrals, time-efficient service, teamwork, reminders); content of care (counseling/

(a)

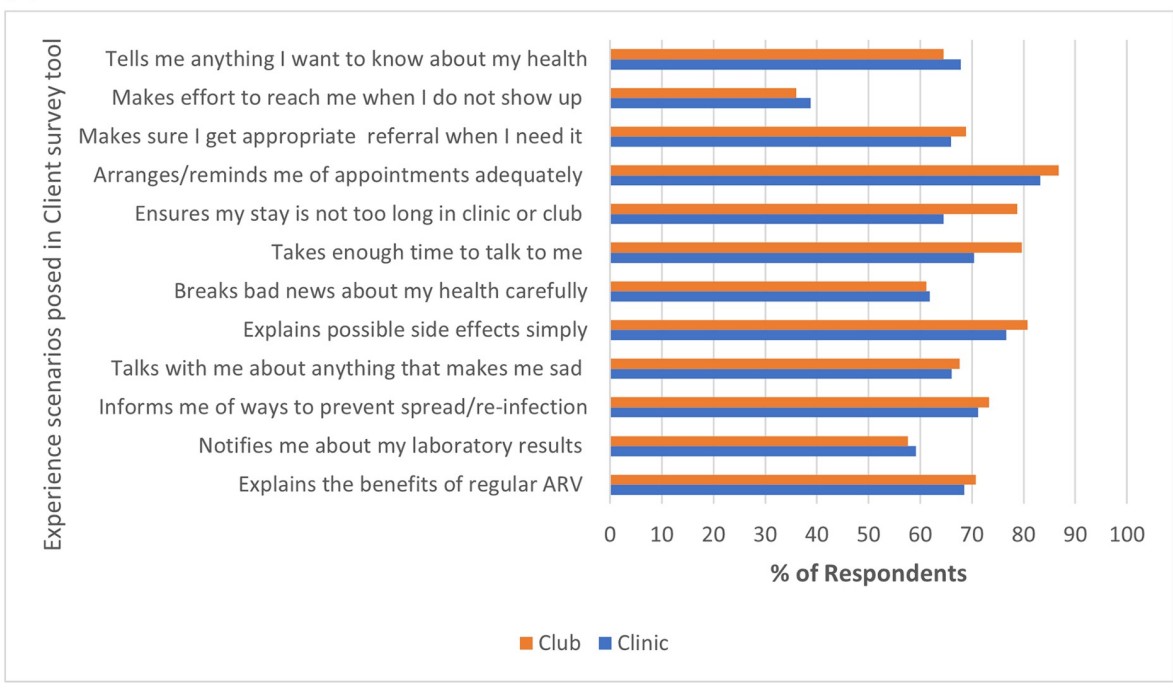

(b)

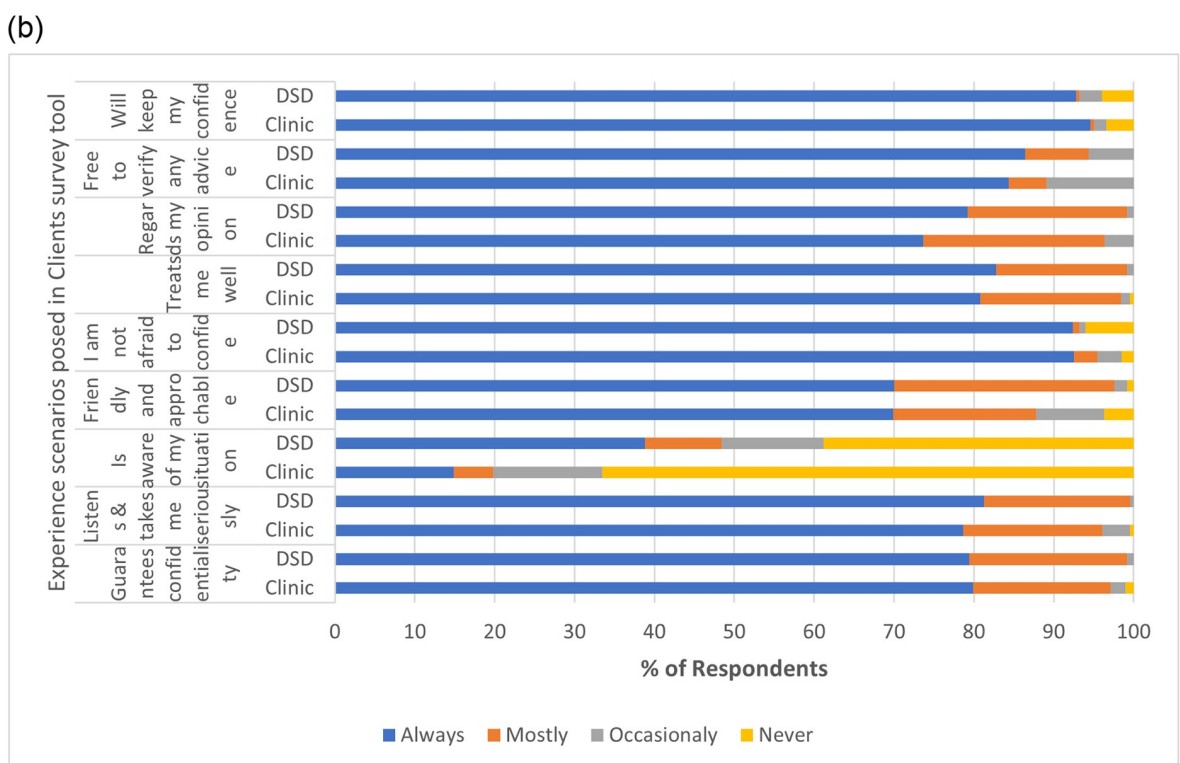

**Fig 2.** **a**: Process of care: Clients' perspective. **b**: Outcome of care: Clients' perspective.

**Table 4. Process of care: Clients perspective and HCW care delivery.**

| | Clinic (N = 378) | Club N = 251 | p value |
|---|---|---|---|
| ***Activity completed during all last 3 visits/meetings- Extracted data from client folder n,%*** | | | |
| Weight taken | 339, 89.7 | 213, 84.9 | 0.071 |
| Screened for OI and TB | 323, 85.4 | 197, 78.5 | 0.024 |
| *Referrals for O1 or TB documented as done | N/A | 63/68, 92.6 | |
| ARV dispensed | 370, 97.9 | 236, 94.0 | 0.012 |
| Adherence assessed | 344, 91.0 | 236, 94.0 | 0.167 |
| ***Client survey*** | | | |
| Respectful service | 376, 99.5 | 251, 100 | 0.248 |
| Recent VL test done (i.e., ≤12 months) | 375, 99.2 | 245, 97.6 | 0.081 |
| Time spent during visit in minutes–*(Median, IQR)* | 119.9, 75.0–180 | 49.9, 33.3–76.6 | <0.001 |

ARV—Antiretroviral drug; IQR—Interquartile range; OI—Opportunistic Infections; TB—Tuberculosis; VL—Viral load

*Clinic participants see the clinician during visit while DSD participants are referred to the clinic; Club % is among those referred

information sharing, health monitoring screening); and manner of care provision (individual attention, non-discriminatory service, respect).

*Practice of care*: Clients associated reduced waiting time for services with quality of care. The club model reduced waiting time both in the clinics and clubs, but clients felt that sufficient time is spent to attend to sick clients.

> "quality service is when we come here we don't even spend an hour if there are some sick people then we get divided by the doctor that for those with no problem should go there get the medicines and leave but for those with problems have to see the doctor go to the laboratory maybe but it's not the same for us who just come and leave after half an hour so I am happy for that, some other places you can stay for a long time, even losing hope to get service"
>
> —*Male clinic participant*

*Content of care*: Clients considered the counseling, health information, and screenings they received as crucial in defining QoC, but they emphasized that showing empathy on other life issues, not necessarily HIV-related also shapes their perception of care received.

> "On average, their talks are good. They are using convincing words which strengthen us, also their deeds are good"
>
> —*Male clinic participant*

*Manner of care provision*: Relatedly, clients also expressed being served respectfully, non-discriminatory service, flexible services, and receiving reminders for appointments including laboratory tests, via call or text message from HCW or through fellow club members as important attributes of quality care. While all clients admitted to receiving respectful service, Club participants in retrospect emphasized a difference between just doing the assigned tasks and actual care.

"Also, when I was going to [clinic name] there were some HCW. they were not polite/humble they seemed as they are fulfilling their responsibilities, which is different from these people who are coming to us nowadays."

*Male club participant*

**Healthcare workers.** *Quantitative.* The routine practice of care by HCW as extracted from the documentation in client folders revealed that ARVS were dispensed for more clinic clients i.e. 98 vs 94% during the last 3 clinic visits/ club meetings, while >90% of clients in clubs got a referral to the clinic who needed (see Table 4).

There were no differences in weight measurements or assessment of adherence. Routine screening for opportunistic infections and TB was documented for more clinic clients i.e.85 vs 79% (see Table 4).

There were no indicators routinely collected for assessing how care was provided by HCW at the time of data collection, therefore, we present only results obtained through interviews in the section below.

*Qualitative. Practice of care*: Time-efficient care provision—HCW reported more time to spend on providing care and conducting daily routines due to reduced client numbers in the clinic. Less work pressure also meant making fewer mistakes with medication.

"It is good. It helps to reduce the population to the clinic, reduce the population you know so make it easier for the workers to work well because sometimes previously it was full, full of clients at the clinic maybe people would start working from morning to six pm. They have worked tiredly; they can't give the right doses they lie to medication sometimes they become unclear they didn't work well. But now they are working well. They enjoy working at the clinic"

*Female club HCW* about the clinic.

Teamwork:—Both clinic and club staff considered the availability of different HCW cadres equipped with basic tools (equipment, tests, medicines) who work as a team as essential for providing quality care.

"If they work like a team then they will provide quality services because when working in team there is cooperation starting with the HBC to the doctor, nurses and the pharmacist if the services are done on time then the patient will be happy and will be of quality"

*Male clinic HCW*

Club staff additionally mentioned that fewer clients miss appointments due to time-efficient services in the clubs.

Non-discriminatory service—HCW in the clinic emphasized the need to talk without using abusive or judgmental words. HCW also emphasized the importance of the environment of the client and tailoring services accordingly.

"The way they welcome him [the client], talk to him. . .. without being abused, stigmatized. What we say maybe can make the service to be of quality. And the way I welcome people because you are not to punish them nor say something that may annoy him, [like] Where were you? What happened? No, you are to use friendly language and read the environment

of the person you want to serve because if you know what the person wants, then he will give you a chance to serve him the way he wants or likes"

*Male clinic HCW*

### Outcome of care

**Clients.** *Quantitative.* The proportion of participants suspected to have TB, or another opportunistic infection revealed no difference between the clinics and clubs (Table 5). Similarly, no difference was seen in the most recent CD4 count between the groups. Time on ART was significantly longer among club participants (3.45 vs 4.27yrs). The proportion of participants with a recent viral load <50 cells/mm was high in both groups but statistically lower among club participants—see details in Table 5.

The majority of clients were confident about the level of confidentiality of their interactions with their HCW. In the clinic and club alike (94.6 vs 92.8% p = 0.60), participants were sure that their HCW would keep their confidence and so expressed little fear about confiding in their HCW -**see** Fig 2b. In both club and clinic, though only a minority reported that HCW were aware of their situation at home and work (14.2 vs 38.7% p = 0.00), most participants were unanimous in the opinion that HCW were friendly and approachable. Compared to clinic participants, more club participants expressed being taken seriously (78.7 vs 81.3% p = 0.05), their opinion being regarded as serious (73.7 vs 79.2% p = 0.06) in managing their health and to being treated well.

*Qualitative. Peer support and client satisfaction*: Both clinic and club clients as did HCW reiterated the value of peer connection.

"But also, I managed to meet a lot of friends that I didn't know but just met them in the services, so the service connects different people and friends"

*Male clinic participant*

"When you sit here you advise each other and leave happy while everyone with her secrets knowing that we talked so and so even if you leave home with stress when you come to the club and sit with your fellows and discuss other things, you forget about home issues and leave in peace"

*Female club participant*

**Table 5. Outcome of care: Client folder review.**

| | Clinic (N = 378) | Club N = 251 | p value |
|---|---|---|---|
| *Extracted data (Documented in folder during last 3 visits/meetings n,%* | | | |
| OI or TB suspected–*n*, % | 103, 27.3 | 68, 27.1 | 0.96 |
| Visit/Meeting attended *(proxy for retention in care)* | 369, 97.6 | 251, 100 | 0.014 |
| Time on ART—*Median, IQR (years)* | 3.45, 2.08–6.06 | 4.27, 2.24–7.61 | 0.002 |
| Most recent CD4 count—*Median, IQR (cells/mm3)* | 500, 334–500 | 515, 359–747 | 0.332 |
| Most recent VL <50 cells/mm–*Median, IQR* | 0, 0–0 | 0, 0–0 | 0.888 |
| Proportion with recent VL <50cells/mm–*n*, % | 375 (99.2) | 237 (94.4) | <0.001 |

ARV—Antiretroviral drug; HCW—Health Care Workers; IQR—Interquartile range; OI—Opportunistic Infections; TB—Tuberculosis; VL—Viral load

*Cost-saving and keeping appointments*: Club participants emphasized that saving money due to not having to travel to access care is important for quality care. Additionally, no club members had missed any meetings for the three most recent appointments which is a boost for clients' retention in care (see Table 5).

> "Here (club) is very near, there is no cost but there (clinic) we were using money, sometimes you can find that I don't have money"
>
> —*Female club participant*

*Improved self-worth due to confidentiality*: An overwhelming majority (>90% in both clinic and club) of clients described keeping their status confidential as an essential element of quality care. Generally, clients associated keeping their status confidential with a sense of self-worth and confidence to face life

> "The secrets are kept so that the community won't discriminate us and we to feel like other people and that it shouldn't discriminate us, we live a good life like others"
>
> —*Female club participant*

*Individual perception of discrimination*: Both clinic and club participants felt that the service models they were engaged in ensured secrecy and therefore prevented discrimination. Interestingly clinic and club participants differed in their assessment of which service-delivery model ensured confidentiality the best. While for clinic participants the clubs represented a risk of unintentional disclosure, Club participants choose the clubs because they felt at risk of unintentionally disclosing their status due to frequent clinic visits as described below:

> "I think I will not join a club because in villages they are advertising to other people that is why I'm fearful about this. I will be here at [the clinic] because I like the services I'm receiving here. In villages there is no secret, people will talk more about me.
>
> —*Male clinic participant*

> "There is no person who is talking in the street about what takes place here. There are many advantages [to the club] for keeping [your status] secret because when we were going to the clinic people were often talking. They said: "Do you see those people? They are going to take drugs". But now they think we are attending a normal meeting because this is an HBC's [Home Based Care worker's] house. And the HBC is a street leader, so they think we are having a normal meeting.
>
> —*Female club participant*

**Healthcare workers.** As outcomes of care are only applicable to clients, we only present the results of the interviews with HCW expressing their perspective about the outcomes of care they provided below.

*Qualitative*. HCW and clients defined quality care by the direct outcome and/or effect of care provided or perceived care they received. The main outcomes used to describe quality care fell in those two broad themes, namely client-related outcomes (laboratory indices, client satisfaction, cost-saving) and confidence/ improved self-worth (due to feeling safe, seen, and valued, the confidentiality of service delivery).

*Laboratory indices*: HCW stated that they assess the health and progress of clients and use that as a measure of the QoC they provide i.e., good progress equates to good care and vice versa. In doing so they use measures such as CD4 count and VL.

> "If I take all the tests required if it's the CD4 and find them high I will be happy and know that I provide quality service and they understand my health training and properly take their medicines or if I test for HVL and find them low if they were 20 and I find are undetectable then I know that I provide quality services"

*Female club HCW*

*Client satisfaction*: HCW also named subjective outcomes such as client satisfaction revealed by the outward expression of happiness.

*Confidentiality/Improved self-worth*: HCW also emphasized confidentiality as a core part of what entailed quality care and had strong ideas about how confidentiality should be ensured. This included strict compliance with professional work ethics which binds them to maintain client confidentiality, which they all emphasized they adhered to. They described ways to accomplish this which included not discussing client information with others, attending to clients one by one in a confidential space to allow privacy. Also, in documenting information in registers and client folders, HCW emphasized the necessity to store these in secure places accessible to HCW alone. Documenting such information using codes that are only understandable to HCW to further ensure confidentiality and for client information entered into the electronic data platform restricted by passwords known only to authorized HCW.

> "the client's information remains confidential because we always do the same work, not everyone is able and that's why we use codes or something that for the ones not involved may not understand that's why if you look at this paper or medical prescription, we don't write the names we write numbers, yes the names are there but when registering these we use codes so I as the professional it makes me secretive"

*Male clinic HCW*

The main quantitative and qualitative findings comparing the quality of care between the clinic and club from the perspective of clients and HCW are summarized in the joint display table (Table 6).

## Discussion

This study evaluated the QoC in ART clubs compared with standard clinic care employing the Donabedian classic framework as a guide. We enabled in-depth understanding by eliciting the perspectives of clients and HCW on the subject. Our results revealed the non-inferiority of QoC provided in clubs when compared with standard care in clinics. Relatedly, clients and HCW alike considered HIV treatment services of good quality irrespective of the service delivery model. Among clients, process-related themes e.g., time spent, confidentiality, and respect were the most important emphasized for describing quality care in both clinic and club. HCW perspectives, on the other hand, emphasized structure-related themes such as availability of resources, decongestion of the clinic, and hence more time for other duties. Both clients and HCW describe quality in terms of outcomes using similar themes such as costs saved (time and money), client's health progress, and expressed satisfaction. No large contradictions were found between quantitative and qualitative findings, qualitative findings added depth and understanding to the quantitative picture.

**Table 6. Joint display table summarizing quantitative and qualitative results by structure, process, and outcome.**

| | lients | | HCW | |
|---|---|---|---|---|
| **Quantitative** | **Qualitative** | **Quantitative** | **Qualitative** | |
| **Structure of care** | | | | |
| • No difference in care experience, except club HCW were better reachable by mobile phone<br>• In both clinic and club, only 70% reported HCW ensured regular ART supply<br>• The service delivery venue was perceived to be inadequate in half of clubs versus almost none in clinics | • Clients defined QoC, differently for clubs (ease of access) and for clinics (centers of expertise).<br>• QoC was associated with adjunct non-medical services.<br>• Clubs led to improved QoC via perceived decongestion of clinic<br>• Choice of club venue was crucial in maintaining confidentiality and to prevent unwanted disclosure. Clubs could be hosted in clinic-spaces or in a village space where it was normal for groups to meet.<br>• Majority of male clients preferred clinic to club-based care. | • Half of club staff found the location to be inadequate, versus 94% of the clinic staff<br>• High proportion of staff in both clinic and club were adequately trained for their tasks<br>• Clinic staff often had other tasks than HIV care, but did not think QoC was affected<br>• Suitable provisions and ART availability were the same in clinics and clubs, half of club HCWs reported inadequate locations for meetings<br>• No differences in logistical, organisational or data managerial aspects between clubs and clinics<br>• Half of clinic staff was unhappy with remuneration versus none of club HCW | • Clinic staff were perceived to have more specialised skills than club HCW<br>• Club and clinic HCW had equal ability to ascertain eligibility criteria, follow guidelines. QoC was seen as ability to maintain ordered documentation.<br>• Clinics were decongested by the clubs, in clubs however need for more permanent location<br>• Both clinic and club HCW perceived support to be adequate, increased pay and training opportunities would lead to better motivation. | |
| **Process of care** | | | | |
| • Care in clubs was considered more timesaving than in clinics<br>• Time spent in the clinic was over double the time spent in the club<br>• HCW in clubs had more time for clients than in clinics<br>• There was no difference in information provision or required procedures performed during a visit | • Both clients in clubs and clinics felt they were given enough time for consultation<br>• Enquiries into broader life areas than just HIV were considered QoC<br>• clients valued flexibility, respect and reminders by HCW or fellow club members as central to QoC. Club participants perceived care in clubs as more emphatic than in the clinic | • A slightly lower proportion was dispensed ART in clubs than clinics<br>• Over 90% of club participants who needed a clinic referral based on guidelines were actually referred<br>• Less routine screening for opportunistic infections was performed in clubs compared to clinics | • HCW reported more time for consultations and lower work-pressure, therefore less mistakes.<br>• Time-efficiency led to fewer missed appointments in clubs.<br>• HCW felt smooth teamwork between different cadres of staff ensured QoC<br>• -HCW perceived QoC as looking at the broader life circumstances of clients. | |
| **Outcome of care** | | | | |
| • There was very high retention in care in both clubs and clinics<br>• Participants in clubs had been on ART for longer than in clinics<br>• Viral suppression proportions were high, although lower among club than clinic participants<br>• Clients had high confidence in confidentiality of HCW in both clinics and clubs<br>• Clients felt HCW in clubs were more aware of clients' home situation than HCW in clinics, and they felt taken more seriously in clubs | • Both clinic and club participants valued the peer networks that had emerged.<br>• Club members emphasized reduction of travel costs and time as QoC.<br>• Clubs facilitated adherence.<br>• Keeping status confidential was a core aspect of QoC and led to improved self-worth.<br>• Both the clinic and the club model ensured confidentiality but in different ways. This perception shaped participant's choices for clinic or club. | • Not applicable | • HCW use measures such as CD4 and VL as indicators of whether they provide QoC<br>• HCW strongly associated QoC to ensuring client's confidentiality. This pertained to all practices from not discussing a client, to seeing clients in a private space to secure storage and coding of documents. | |

In terms of the structure of care, our finding that the availability of resources was similarly important for clinic and club participants and led to a positive perception of quality among clients and HCW alike is intuitive and aligns with the literature [22, 23]. HCW emphasized structure-related attributes e.g., equipment, information management systems, more than clients. Even with reasonable support structures in our study setting, irrespective of delivery model,

the suggestions for more training opportunities, and better remuneration among HCW were seen as improving QoC even with other DSD types [24–26]. The availability of HCW as a measure of quality care was cited mainly by clinic participants, non-mention among club participants likely reflects that these were taken for granted rather than are not important. The shortage of HCW leading to long waiting times in facilities is an identified reason for client disengagement from service and motivation for DSD [24]. Similarly, the reliable supply of basic commodities e.g., ARVs, laboratory tests, etc. considered essential by other DSD studies for the provision of quality care, was also found in our study [27–29].

While some studies report the reluctance of clients to participate in DSD due to perceived higher quality of clinic-based care, others support our finding that clients considered DSD desirable and convenient [27, 28]. Our finding that the provision of adjunct non-medical services such as breakfast, food items, and monetary aid was a major influence on the perception of quality is likely associated with the prevailing poverty in our study setting. While this suggests adjunct interventions that can be considered alongside DSD, it raises concerns about sustainability. Additionally, having a conducive and confidential space for service delivery was echoed in our study as an important aspect of quality which is consistent with other studies where clients cited the fear of unintentional disclosure as a reason for not keeping clinic appointments [8, 24]. Our study revealed that what constitutes a confidential space is individually perceived e.g., for clinic participants, it referred to receiving care outside of the social control in the village, whereas for club participants not having to go to a place known as dispensing drugs for HIV was a way to maintain quality care. Moreover, venues selected by clients for ART club were normal meeting places for several groups in order not to arouse any suspicions which could fuel stigma. This alludes to the minimal impact of DSD on HIV-related stigma, in agreement with current evidence in the literature [30–32]. The better funding of the study sites, when compared to public facilities in terms of staff, equipment, and commodities being owned by the Catholic mission and supported by the T&T project, may partly explain our findings. Moreover, clubs were perceived as extensions of the clinics and very much connected.

Whereas HCW focused more on structure as a key aspect of QoC, clients were more concerned with processes. Key care processes were similarly conducted between clinic and club, with clubs revealing better time efficiency. Timely service delivery as a key aspect of quality care resonated throughout our study and is corroborated across many DSD studies as a core part of quality care for clients and HCW [28, 29, 33, 34]. While club clients spent the least amount of time, both club and clinic participants agreed that timely service had improved with the introduction of the DSD model. Our finding that information sharing and counseling from HCW and also from peers was considered a measure of good quality is coherent with the literature [31]. Interpersonal relations with HCW and personable services delivered in a culturally acceptable way and with attention to broader life circumstances of the client, were major influences shaping the perspectives of clients e.g., respectful service delivery ranked highly in the consideration of quality and is consistent with findings from other studies [31, 35]. Our study observed a particular coherence in views of clients and HCW around respect as part of quality, a finding that differs from another related study [36]. This may be related to the prevailing cultural environment which extols politeness and respect as a way of life [24]. It is noteworthy that studies are showing poor HCW attitude as contributing to client drop-out from ART care [37–39]. The optimal attendance we found supports this and therefore suggests? that the club model may facilitate retention among clients who would otherwise be lost-to-follow-up if extended to clients not considered stable. Relatedly, our finding that improved individualized and non-discriminatory care was perceived favorably aligns with a core motivation for DSD namely, client-centered care [31].

Expectedly, core outcomes of QoC such as the most recent CD4 count, and viral load results were similar as our comparison was only among stable clients. Similarities in recent viral load and CD4 count between clinic and club affirms findings from other studies that DSD does not threaten the attainment of desired treatment outcomes [6, 40]. While we found no difference between clinic and club in the proportion of clients with a recent viral load test (<12 months, according to national guidelines), the lower proportion of club clients with results <50 copies/ml may be explained by the few clients who were allowed to participate in clubs with good adherence and a VL result between 50–200 copies/ml (considered as virologic blips) [41]. It may also reflect the early phase of club implementation which requires a period of familiarization with the protocol. Continuous mentoring and monitoring of adherence to procedures by HCW is advocated as in other DSD interventions [29, 42].

While HCW emphasized improved client status, clients emphasized improved social standing and peer support as outcomes of care. Regaining a sense of self-worth and a more positive outlook on life was an important outcome influencing perceptions of HIV care quality in our study and was also seen in other studies [30, 43]. This is closely linked to improved health status, to which ART no doubt contributes to a large extent. Evidence shows that stigma remains an issue in our study setting as in many others. The fear of unintentional disclosure may likely account for the import placed on the assurance of status confidentiality as a measure of quality [43–46]. Additionally, our finding that valuing clients' opinions empowers them to self-manage and feel like allies align with the literature [47, 48]. Beyond psychosocial support, initiatives to support financial wellbeing ranked high, especially among club participants. The prevailing socio-economic conditions in our setting i.e., poverty, illiteracy, and dependence on subsistence farming as major income earners likely explain this finding. The literature shows that DSD offers convenience while saving costs (time and money) which underlies its acceptability across settings [8, 28, 34, 48, 49]. Not surprisingly, savings in terms of reduced access cost (transportation) was mentioned mainly among club participants.

## Strength & limitations

The Donabedian framework facilitated the vigorous assessment of QoC taking into consideration that while some aspects of quality care are easy to measure, others are not so. The interpersonal process which drives interaction between care recipients and providers complicates the assessment of QoC. Utilizing the three-dimensional framework enabled the broad exploration of QoC in our study, especially as the relationship between structure, process, and outcome are considered non-linear [13, 36]. We employed a mixed-method study design triangulating quantitative and qualitative data to provide more valid evidence. Our findings must be interpreted in the face of some limitations. The study was conducted one year following the implementation of clubs in the study clinics to provide the intervention with data to improve the model. This means that we were not able to measure the long-term impact of the DSD model on QoC. Our findings however mostly align with the literature even for similar interventions that have been implemented for longer [9, 29, 32, 50]. Moreover, our study sites were mission-owned facilities that are generally better funded and may likely not represent the situation in publicly owned hospitals. It is also possible that some participants gave socially desirable responses. The tendency of clients to refrain from saying disagreeable statements may have influenced responses. We countered this by training research assistants to assure participants that their responses would be anonymized and have no effect on the services they receive but rather provide ways for improving services. There may have been selection bias i.e., those who opted to join clubs may have had different attitudes, status geographic considerations, etc., to those happy to stay at the clinic. By allowing clients' preferences to determine

which service delivery model they receive, we believe our findings are relevant to what can be expected in real life.

## Conclusion

Our study revealed that QoC was comparable between the ART club and the clinic in our setting in Shinyanga, Tanzania. The perspectives of clients and HCW complemented quantitative results consistently across the sub-dimensions of structure, process, and outcome of care. While HCW emphasizes structure in defining QoC, clients focus on process. The need for both clinic-based care and DSD was apparent as many elements of quality care were individually perceived. Regardless of the model/type of care, QoC will benefit from other interventions addressing socio-economic situations of widespread poverty such as we have in our study setting. Ultimately, contextually relevant adaptations informed by the perspectives of clients and HCW alike will be important for achieving acceptable QoC in DSD interventions across settings. Future research can investigate the longitudinal evaluation of outcomes of care and explore employing ethnographic studies to delve deeper into how the DSD model is perceived over time to inform relevant adaptations in service delivery.

## Supporting information

**S1 Appendix. Quality of care questionnaire for patients-English.**
(PDF)

**S2 Appendix. Quality of care questionnaire for patients-Swahili.**
(PDF)

## Acknowledgments

The implementation of the Shinyanga and Simiyu Test & Treat program in Tanzania is by Doctors with Africa CUAMM and the Diocese of Shinyanga within the framework set out in the national guidelines of the Tanzanian Ministry of Health, Community Development, Gender, Elderly and Children (MoHCDGEC) through the National AIDS Control Program (NACP). The scientific evaluation of the project is under the guidance of Principal Investigators Prof Anton Pozniak and Dr. Bernard Desderius and is performed by the Amsterdam Institute for Global Health and Development (AIGHD) in collaboration with Doctors with Africa CUAMM. The content of this manuscript is solely the responsibility of the authors and does not necessarily represent the official views of any of the institutions mentioned above. We thank all our institutional collaborators, the study participants, the staff at the project clinical sites and laboratories, as well as the project support staff for their invaluable support to this program in general and the current manuscript in particular.

## Author Contributions

**Conceptualization:** Nwanneka Ebelechukwu Okere, Denise Naniche, Gabriela B. Gomez, Tobias Rinke de Wit, Josien de Klerk, Sabine Hermans.

**Data curation:** Nwanneka Ebelechukwu Okere, Josien de Klerk.

**Formal analysis:** Nwanneka Ebelechukwu Okere, Judith Meta, Josien de Klerk, Sabine Hermans.

**Funding acquisition:** Tobias Rinke de Wit.

**Investigation:** Nwanneka Ebelechukwu Okere, Judith Meta.

**Methodology:** Nwanneka Ebelechukwu Okere, Josien de Klerk, Sabine Hermans.

**Project administration:** Nwanneka Ebelechukwu Okere.

**Supervision:** Denise Naniche, Gabriela B. Gomez, Tobias Rinke de Wit, Josien de Klerk, Sabine Hermans.

**Validation:** Judith Meta, Giulia Martelli, Josien de Klerk, Sabine Hermans.

**Visualization:** Nwanneka Ebelechukwu Okere.

**Writing – original draft:** Nwanneka Ebelechukwu Okere, Judith Meta, Josien de Klerk, Sabine Hermans.

**Writing – review & editing:** Nwanneka Ebelechukwu Okere, Judith Meta, Werner Maokola, Giulia Martelli, Eric van Praag, Denise Naniche, Gabriela B. Gomez, Anton Pozniak, Tobias Rinke de Wit, Josien de Klerk, Sabine Hermans.

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
