## [Decision Letter · Decision Letter 0]

22 Jun 2021

PONE-D-20-41086

Quality of care in a differentiated HIV service delivery intervention in Tanzania: A mixed methods study

PLOS ONE

Dear Dr. Okere,

Thank you for submitting your manuscript to PLOS ONE. After careful consideration, we feel that it has merit but does not fully meet PLOS ONE’s publication criteria as it currently stands. Therefore, we invite you to submit a revised version of the manuscript that addresses the points raised during the review process.

Thank you for the opportunity to serve as academic editor on this manuscript. Thank you again for your patience regarding this manuscript. We battled to get  reviewers to review this manuscript with as many as five reviewers invited declining to review the manuscript. I eventually got a good review and have also reviewed the manuscript carefully myself. 

We look forward to receiving your revised manuscript.

Kind regards,

Tendesayi Kufa, MBChB, PhD

Academic Editor

PLOS ONE

Journal Requirements:

2. Please include a Swahili copy of the questionnaire, as Supporting Information, or include a citation if it has been published previously.”

3. In statistical methods, please refer to any post-hoc corrections to correct for multiple comparisons during your statistical analyses. If these were not performed please justify the reasons. Please refer to our statistical reporting guidelines for assistance (https://journals.plos.org/plosone/s/submission-guidelines.#loc-statistical-reporting).

"The Shinyanga and Simiyu Test & Treat program in Tanzania is supported by Gilead Sciences (USA) and the Diocese of Shinyanga through the Good Samaritan Foundation (Vatican). The

implementation of the project is by Doctors with Africa CUAMM and the Diocese of Shinyanga within the framework set out in the national guidance of the Tanzanian Ministry of Health, Community Development, Gender, Elderly and Children (MoHCDGEC) through the National AIDS Control Program (NACP). The scientific evaluation of the project is under the guidance of Principal Investigators Prof Anton Pozniak and Dr Bernard Desderius and is performed by the Amsterdam Institute for Global Health and Development (AIGHD) in collaboration with Doctors with Africa CUAMM. The content of this manuscript is solely the responsibility of the authors and does not necessarily represent the official views of any of the institutions mentioned above. We thank all our institutional collaborators, the study participants, the staff at the project clinical sites and laboratories, as well as the project support staff for their invaluable support to this program in general and the current manuscript in particular."

"The Shinyanga T&T project is funded by Gilead Sciences Inc. ONE was funded by the

Erasmus Mundus Joint Doctorate Trans Global Health Programme EMJD-TGH

(Framework Partnership Agreement 2013-0039, Specific Grant Agreement 2014-0681)

http://www.transglobalhealth.org/

and the Amsterdam Institute for Global health and Development (AIGHD). The funders

had no role in study design, data collection and analysis, decision to publish or

preparation of the manuscript."

Additionally, because some of your funding information pertains to commercial funding, we ask you to provide an updated Competing Interests statement, declaring all sources of commercial funding.

In your Competing Interests statement, please confirm that your commercial funding does not alter your adherence to PLOS ONE Editorial policies and criteria by including the following statement: "This does not alter our adherence to PLOS ONE policies on sharing data and materials.” as detailed online in our guide for authors  http://journals.plos.org/plosone/s/competing-interests.  If this statement is not true and your adherence to PLOS policies on sharing data and materials is altered, please explain how.

Please include the updated Competing Interests Statement and Funding Statement in your cover letter. We will change the online submission form on your behalf.

Additional Editor Comments (if provided):

Generally I found the manuscript well written. It was however long and could be improved in some parts.  Please see more detailed comments below. 

Abstract

• Line 30: Shouldn’t it be QoC in ART clubs vs clinics specifically and not DSD in general? My understanding is that DSD goes beyond ART clubs and encompasses MMD, fast track visits, drug delivery etc.

• Lines 40- 41: Did the authors look at VL completion or retention in care as an outcome?

Methods

• Line 120- 124: Why was retention in care used as a proxy for QoC? How was this retention defined? Provide justification

• Line 138: add a section which looks at DSD in the clinic setting. What did it look like for stable on ART patients who are otherwise eligible for ART clubs but opt not to go to clubs? What were the guidelines for managing stable on ART clients in the clinic

• Lines 140- 144: was this modified tool validated?

• Lines 159 – 163: the authors refer to core outcomes? Shouldn’t this be primary or main? Also the study was powered on retention but yet this was not determined as an outcome in the study. Why?

Results

As expected of a mixed method study, this section is quite long and could have flowed better by separating patients/ client issues from HCW issues. Consider presenting structure, process and outcomes sections separately for patients/ clients then for HCWs

• Line 192: In conversation? Meaning here is uncertain

• Why tables 1a/1b when tables are presenting information independent of each other. Perhaps just tables 1 or 2 and include others as supplementary tables if needed

• Lines 248- 251: What is meant by training opportunities for staff? By self –development opportunities, you mean lack of career progression with staff feeling stuck doing only HIV

• Lines 325- 326: Is it possible to quantify in terms of percentages the greater likelihood by males to opt to remain in clinic based care as opposed going into ART clubs

• Line 405 – Is it possible to call this something else. It’s similar to an earlier section in line 338

• Table 2c – Did the authors consider retention in care and VL completion as outcomes

Discussion

• Lines 622 - ? DSD vs ART clubs

• Lines 633- 635: Could ART clubs worsen stigma? If an important component of stigma reduction is “normalizing” HIV and HIV care, ART clubs may “exceptionalise” HIV and make the perception and experience of stigma worse.

• Lines 639: How did you arrive at this conclusion? Was this well highlighted in the results?

Reviewers' comments:

Reviewer's Responses to Questions

**Comments to the Author**

1. Is the manuscript technically sound, and do the data support the conclusions?

Reviewer #1: Partly

2. Has the statistical analysis been performed appropriately and rigorously? 

Reviewer #1: Yes

3. Have the authors made all data underlying the findings in their manuscript fully available?

Reviewer #1: Yes

4. Is the manuscript presented in an intelligible fashion and written in standard English?

Reviewer #1: No

5. Review Comments to the Author

Reviewer #1: Thank you for the opportunity to review your manuscript. See below comments/suggestions

1. Minor: Patients eligibility criteria – Is it correct that the patients should have attended at least 3 club sessions? I might have missed this, but other than being stable on ART, there appears to be no mention of other eligibility criteria for the club and clinic patients

2. Minor: Age bands in Table 1 appear to be overlapping – i.e., the upper category of the range is included as the lower value of the next range. For example, it appears that “35” is included in both the second and third age categories

3. Minor: Sentence on 213 needs to be revised for clarity

4. Minor: Sentence 215 – did the FGD participants also participate in the in-depth interviews. If so, how was this handled in analysis.

5. Major: General comment – the use of mixed methods has its strengths. However, the inclusion of all the findings in one manuscript loses the reader and makes the paper difficult to follow. In addition to there being multiple dimensions on QoC, the reader also needs to (1) contend with comparisons between clinic and club participants, and (2) also incorporate the qualitative and quantitative data strands

6. PLOS authors have the option to publish the peer review history of their article (what does this mean?). If published, this will include your full peer review and any attached files.

Reviewer #1: No

---

## [Author Response · Author response to Decision Letter 0]

4 Aug 2021

Response to Reviewers

Journal Requirements:

 The manuscript and author affiliations have been formatted according to PLOS ONE’s style requirements

2. Please include a Swahili copy of the questionnaire, as Supporting Information, or include a citation if it has been published previously.”

 The Swahili copy of the questionnaire has been included as a supplementary file (see S1 Appendix)

3. In statistical methods, please refer to any posthoc corrections to correct for multiple comparisons during your statistical analyses. If these were not performed please justify the reasons. Please refer to our statistical reporting guidelines for assistance (https://journals.plos.org/plosone/s/submission-guidelines.#loc-statistical-reporting).

We have now employed the Bonferroni correction in our study to guide the interpretation of which comparisons were reported as significant. We have included this in the Data Analysis and Results sections, see lines 166-168 and 239-240 respectively. 

There must have been some miscommunication, as the data used for our study are freely available online. They can be accessed at https://datadryad.org/stash/dataset/doi%253A10.5061%252Fdryad.bcc2fqzbj

The qualitative data contained potentially identifying participants' information and were not included in the dryad dataset. The data can be accessed upon reasonable request from the Shinyanga T & T project scientific committee through secretariat@aighd.org

" The Shinyanga and Simiyu Test & Treat program in Tanzania is supported by Gilead Sciences (USA) and the Diocese of Shinyanga through the Good Samaritan Foundation (Vatican). The implementation of the project is by Doctors with Africa CUAMM and the Diocese of Shinyanga within the framework set out in the national guidance of the Tanzanian Ministry of Health, Community Development, Gender, Elderly and Children (MoHCDGEC) through the National AIDS Control Program (NACP). The scientific evaluation of the project is under the guidance of Principal Investigators Prof Anton Pozniak and Dr. Bernard Desderius and is performed by the Amsterdam Institute for Global Health and Development (AIGHD) in collaboration with Doctors with Africa CUAMM. The content of this manuscript is solely the responsibility of the authors and does not necessarily represent the official views of any of the institutions mentioned above. We thank all our institutional collaborators, the study participants, the staff at the project clinical sites and laboratories, as well as the project support staff for their invaluable support to this program in general and the current manuscript in particular."

The Acknowledgement section has been revised by removing the funding information contained in the first statement. It now reads;

" The implementation of the Shinyanga and Simiyu Test & Treat program in Tanzania is by Doctors with Africa CUAMM and the Diocese of Shinyanga within the framework set out in the national guidance of the Tanzanian Ministry of Health, Community Development, Gender, Elderly and Children (MoHCDGEC) through the National AIDS Control Program (NACP). The scientific evaluation of the project is under the guidance of Principal Investigators Prof Anton Pozniak and Dr. Bernard Desderius and is performed by the Amsterdam Institute for Global Health and Development (AIGHD) in collaboration with Doctors with Africa CUAMM. The content of this manuscript is solely the responsibility of the authors and does not necessarily represent the official views of any of the institutions mentioned above. We thank all our institutional collaborators, the study participants, the staff at the project clinical sites and laboratories, as well as the project support staff for their invaluable support to this program in general and the current manuscript in particular."

“The Shinyanga T&T project is funded by Gilead Sciences Inc. ONE was funded by the Erasmus Mundus Joint Doctorate Trans Global Health Programme EMJD-TGH (Framework Partnership Agreement 2013-0039, Specific Grant Agreement 2014-0681) http://www.transglobalhealth.org/

and the Amsterdam Institute for Global Health and Development (AIGHD). The funders had no role in study design, data collection, and analysis, decision to publish, or preparation of the manuscript.”

The funding statement has been updated to read thus;

“The Shinyanga and Simiyu Test & Treat program in Tanzania is funded by Gilead Sciences (USA) and the Diocese of Shinyanga through the Good Samaritan Foundation (Vatican). ONE was funded by the

Erasmus Mundus Joint Doctorate Trans Global Health Programme EMJD-TGH

(Framework Partnership Agreement 2013-0039, Specific Grant Agreement 2014-0681)

http://www.transglobalhealth.org/ and the Amsterdam Institute for Global Health and Development (AIGHD). The funders had no role in study design, data collection, and analysis, decision to publish or

preparation of the manuscript."

Additionally, because some of your funding information pertains to commercial funding, we ask you to provide an updated Competing Interests statement, declaring all sources of commercial funding.

In your Competing Interests statement, please confirm that your commercial funding does not alter your adherence to PLOS ONE Editorial policies and criteria by including the following statement: "This does not alter our adherence to PLOS ONE policies on sharing data and materials.” as detailed online in our guide for authors http://journals.plos.org/plosone/s/competing-interests. If this statement is not true and your adherence to PLOS policies on sharing data and materials is altered, please explain how.

 The competing interest statement has been updated to read thus;

“The Shinyanga and Simiyu Test & Treat program in Tanzania is funded by Gilead Sciences (USA). This does not alter our adherence to PLOS ONE policies on sharing data and materials.

[GBG is currently employed by Sanofi Pasteur. Sanofi Pasteur was not involved in any way and did not provide funding for this study]. All other authors declare that they have no competing interests.”

Please include the updated Competing Interests Statement and Funding Statement in your cover letter. We will change the online submission form on your behalf.

 The updated Competing Interests Statement and Funding Statement are included in the cover letter 

Additional Editor Comments (if provided):

Generally, I found the manuscript well written. It was however long and could be improved in some parts. Please see more detailed comments below. 

Abstract

• Line 30: Shouldn’t it be QoC in ART clubs vs clinics specifically and not DSD in general? My understanding is that DSD goes beyond ART clubs and encompasses MMD, fast track visits, drug delivery, etc.

The sentence has been revised by replacing ‘DSD’ with ‘ART clubs’.

• Lines 40- 41: Did the authors look at VL completion or retention in care as an outcome?

Yes. We have revised the abstract to reflect this

You can find the detailed results in Tables 3b & 3c i.e. (lines 427 & 515)

a. VL completion – ‘Recent VL test done’ (% of clients with a VL test done <12 months) 

b. Retention – ‘Visit/meeting attended’ (% of clients who attended the last 3 visits/meetings). We used ‘visit/meeting attendance’ as a proxy for retention to show consistency in care-seeking which is at the core of retention because clubs had just been rolled out about one year and was too soon to measure retention (see lines 126-128).

Methods

• Line 120- 124: Why was retention in care used as a proxy for QoC? How was this retention defined? Provide justification

We have justified using retention in care as a proxy for QoC and also provided the definition we used for retention in our study in lines 123-125

• Line 138: add a section that looks at DSD in the clinic setting. What did it look like for stable on ART patients who are otherwise eligible for ART clubs but opt not to go to clubs? What were the guidelines for managing stable on ART clients in the clinic

We have provided a brief description of a typical clinic visit for stable ART clients at the clinic see 142-145

• Lines 140- 144: was this modified tool validated?

No

• Lines 159 – 163: the authors refer to core (Main) outcomes? Shouldn’t this be primary or main? 

‘Core’ changed to ‘Main’ see line 166

Also, the study was powered on retention but yet this was not determined as an outcome in the study. Why? 

Retention was determined, however, we measured it in a more process-related manner by considering the percentage of clients who attended all their last 3 clinic appointments or club meetings. This was to give a better estimate of clients who would consistently engage in care (assuming that the QoC is a factor contributing to that behavior). We have revised to clarify this - see Table 3c ‘Visit/meeting attended’. We have also included some text explaining this in the methods see lines 123-125 and incorporated it in the abstract (see line 34 ).

Results

As expected of a mixed-method study, this section is quite long and could have flowed better by separating patients/ client issues from HCW issues. Consider presenting structure, process, and outcomes sections separately for patients/ clients then for HCWs

We considered the presentation of our results in many ways including the one suggested by the reviewer. We settled for the current presentation to highlight the interrelatedness of many of the issues raised by both HCW and clients. Apart from minimizing having many sub-headings, we considered that having all the views around a particular topic in one place, rather than having to go back and forth would improve readability. We have however revised the text within each section with clearer signposting to distinguish which findings pertain to HCW and which pertains to clients see lines 239-242, 278, 302-303, 341, 377, 399, 410, 436-437, 472-477, 550-552

• Line 192: In conversation? Meaning here is uncertain

The sentence has been rephrased to improve clarity see line 202

• Why tables 1a/1b when tables are presenting information independent of each other. Perhaps just tables 1 or 2 and include others as supplementary tables if needed

The tables have been renumbered as suggested

• Lines 248- 251: What is meant by training opportunities for staff? By self–development opportunities, you mean lack of career progression with staff feeling stuck doing only HIV

The sentence has been rephrased to clarify meaning see lines 260-261

• Lines 325- 326: Is it possible to quantify in terms of percentages the greater likelihood by males to opt to remain in clinic-based care as opposed to going into ART clubs

The percentages have been included in the text as highlighted in Table 2 see lines 336-337

• Line 405 – Is it possible to call this something else. It’s similar to an earlier section in line 338

c. The sub-section has been renamed appropriately

• Table 2c – Did the authors consider retention in care and VL completion as outcomes

a. Yes, we did. See our response to a similar query above in the fourth bullet point under Methods. We looked at retention more from the angle of a process and therefore measured the frequency of attendance of the last 3 visits which we have now presented in Table 3c) – see ‘Visit/Meeting attended’. As explained above, this variable measured the percentage of patients who attended all last three appointments. We have provided some justification for taking this approach see lines 123-125

b. We considered VL completion in two ways, first as a process – see Table 3b ‘Recent VL test done’ which measured the percentage of patients who had VL test done within 12 months of the data collection date. This was to show the level of adherence to the care process recommended in the treatment guidelines as a measure of quality. Secondly, we considered the VL test result as an outcome and therefore compared the median values between the two groups as well as comparing the percentage of patients who remained virally suppressed – see ‘ Most recent VL <50 cells/mm’ and ‘Proportion with recent VL <50 cells /mm’ in Table 3c

Discussion

• Lines 622 - DSD vs ART clubs?

The sentence was about DSD in general. The studies cited were not ART clubs. The sentence has been revised to clarify this see line 628.

• Lines 633- 635: Could ART clubs worsen stigma? If an important component of stigma reduction is “normalizing” HIV and HIV care, ART clubs may “exceptionalise” HIV and make the perception and experience of stigma worse.

The situation varies with context. Stigma is a complex subject. While we agree that ART clubs may ‘exceptionalise’ HIV and this could further fuel stigma, there were existing social structures in our study setting that were leveraged in setting up the ART clubs in each village e.g. there is home-based care (HBC) staff member, who is lay personnel responsible for health promotion activities in the village. The HBC’s home is the venue for several meetings. ART club members were free to choose the venue for their meeting and in many cases, they preferred the HBC’S home since it was a ‘normal’ meeting place and no one could tell what group was meeting. Otherwise, they could choose wherever they felt safe to meet without discrimination see lines 313-316, and 646-648.

• Lines 639: How did you arrive at this conclusion? Was this well highlighted in the results? 

Yes. We considered the dominant themes in each participant's contributions for each section (i.e. structure, process, and outcome) to draw our conclusion. 

Reviewer #1: Thank you for the opportunity to review your manuscript. See below comments/suggestions

1. Minor: Patients eligibility criteria – Is it correct that the patients should have attended at least 3 club sessions? I might have missed this, but other than being stable on ART, there appears to be no mention of other eligibility criteria for the club and clinic patients

Yes, we targeted clients who have had some experience with the club intervention. For that, we selected clubs that were at least 6 months old.

In Materials and methods under subsection c. (Study sampling procedure), we define stable ART clients used in our study see 116-131

2. Minor: Age bands in Table 1 appear to be overlapping – i.e., the upper category of the range is included as the lower value of the next range. For example, it appears that “35” is included in both the second and third age categories

The age bands have been corrected.

3. Minor: Sentence on 213 needs to be revised for clarity

The sentence has been revised 

4. Minor: Sentence 215 – did the FGD participants also participate in the in-depth interviews. If so, how was this handled in analysis?

The sentence has been revised for clarity. Stable ART clients participated only in FGD, while health care workers were interviewed individually. No one participated in both see lines 175-185

5. Major: General comment – the use of mixed methods has its strengths. However, the inclusion of all the findings in one manuscript loses the reader and makes the paper difficult to follow. In addition to there being multiple dimensions on QoC, the reader also needs to (1) contend with comparisons between the clinic and club participants, and (2) also incorporate the qualitative and quantitative data strands

To enable the exploration of the different aspects and perspectives of Quality of HIV care comprehensively is what motivated the choice of the mixed methods design for our study. We considered sharing our findings in several ways including writing separate articles but we felt that would water down the comprehensive picture we set out to portray. We agree that we share a lot of information which is why we tried to organize our findings per sub-category of structure, process, and outcome. That way, we hope to inform our readers sufficiently about each of the subcategories separately while sharing the perspectives of both clients and HCW from clinics and clubs at the same time.

---

## [Decision Letter · Decision Letter 1]

11 Jan 2022

PONE-D-20-41086R1Quality of care in a differentiated HIV service delivery intervention in Tanzania: A mixed methods studyPLOS ONE

Dear Dr. Okere,

Thank you for submitting your manuscript to PLOS ONE. After careful consideration, we feel that it has merit but does not fully meet PLOS ONE’s publication criteria as it currently stands. Therefore, we invite you to submit a revised version of the manuscript that addresses the points raised during the review process. One of the reviewers has made some valuable point regarding the format of the manuscript and myself as the Academic Editor appreciate their perspective. In this case, all major and minor concerns have been indeed addressed by you, in your rebuttal and the revised manuscript in a satisfactory way. However, I would ask that you make one more effort to increase the readability of your manuscript and improve its formatting to account for the mixed methods approach you have -rightly- undertaken.  I would appreciate if you consult and cite any methodological source(s) in your references. I have personally found particularly helpful most of the methodological papers and manuals by John W Creswell and his team, but there are several others. 

We look forward to receiving your revised manuscript.

Kind regards,

Petros Isaakidis

Academic Editor

PLOS ONE

Journal Requirements:

Reviewers' comments:

Reviewer's Responses to Questions

**Comments to the Author**

1. If the authors have adequately addressed your comments raised in a previous round of review and you feel that this manuscript is now acceptable for publication, you may indicate that here to bypass the “Comments to the Author” section, enter your conflict of interest statement in the “Confidential to Editor” section, and submit your "Accept" recommendation.

Reviewer #1: (No Response)

2. Is the manuscript technically sound, and do the data support the conclusions?

Reviewer #1: Yes

3. Has the statistical analysis been performed appropriately and rigorously? 

Reviewer #1: Yes

4. Have the authors made all data underlying the findings in their manuscript fully available?

Reviewer #1: Yes

5. Is the manuscript presented in an intelligible fashion and written in standard English?

Reviewer #1: No

6. Review Comments to the Author

Reviewer #1: The authors have adequately addressed all minor comments and clarifying questions.

However, while mixed methods approaches bear strengths in triangulating information - careful thought and tact should be given to how the results are presented. We understand that the authors did not want to tone down the findings. However, at the same time - the value of the work seems to be lost in how the results section is currently organised. I would recommend that authors reconsider formatting the results section as suggested by both reviewers. The current revisions made do not adequately address this concern.

7. PLOS authors have the option to publish the peer review history of their article (what does this mean?). If published, this will include your full peer review and any attached files.

Reviewer #1: No

---

## [Author Response · Author response to Decision Letter 1]

25 Feb 2022

One of the reviewers has made some valuable point regarding the format of the manuscript and myself as the Academic Editor appreciate their perspective. In this case, all major and minor concerns have been indeed addressed by you, in your rebuttal and the revised manuscript in a satisfactory way. However, I would ask that you make one more effort to increase the readability of your manuscript and improve its formatting to account for the mixed methods approach you have -rightly- undertaken. I would appreciate if you consult and cite any methodological source(s) in your references. I have personally found particularly helpful most of the methodological papers and manuals by John W Creswell and his team, but there are several others. 

Thank you for another opportunity to revise our paper further and make it more readable. 

1. We have reformatted the entire results section in a bid to improve flow and accentuate the information gleaned from our mixed methods approach. After trying out various options, we decided to maintain the structure of the Donabedian framework – structure, process, and outcome. In each part, first, we present the quantitative findings from clients, followed by their qualitative findings. Then we present the quantitative findings from HCW, followed by their qualitative findings. We repeat the same format for all three parts.

We also renamed the Figures and Tables in an attempt to clarify to which part they belong and tried grouping them as much as possible to improve readability. While revising we came across some minor textual things which we took the liberty of changing; please see the tracked changes manuscript.

2. We have indeed perused the remarkable work done by John W Creswell and a couple of others and based on recommendations for highlighting the main findings for mixed methods research in joint display tables, have created such a table that we have included in our paper, see table 3d. The table summarizes the main findings for the quantitative and qualitative parts. We have included the explanation of the ordering of the results and the inclusion of the joint display table in the methods section, see lines 107-111. We have also included the appropriate citations of the methodology employed.

Reviewer #1: 

The authors have adequately addressed all minor comments and clarifying questions.

However, while mixed methods approaches bear strengths in triangulating information - careful thought and tact should be given to how the results are presented. We understand that the authors did not want to tone down the findings. However, at the same time - the value of the work seems to be lost in how the results section is currently organised. I would recommend that authors reconsider formatting the results section as suggested by both reviewers. The current revisions made do not adequately address this concern.

Thank you for your feedback. We have taken your recommendations and reformatted the entire results section as suggested. We hope that we have adequately addressed your concern by our revision.

---

## [Editor Report · Decision Letter 2]

1 Mar 2022

Quality of care in a differentiated HIV service delivery intervention in Tanzania: A mixed methods study

PONE-D-20-41086R2

Dear Dr. Okere,

We’re pleased to inform you that your manuscript has been judged scientifically suitable for publication and will be formally accepted for publication once it meets all outstanding technical requirements.

Kind regards,

Petros Isaakidis MD, PhD

Academic Editor

PLOS ONE
---

## [Editor Report · Acceptance letter]

4 Mar 2022

PONE-D-20-41086R2 

Quality of care in a differentiated HIV service delivery intervention in Tanzania: A mixed-methods study 

Dear Dr. Okere:

I'm pleased to inform you that your manuscript has been deemed suitable for publication in PLOS ONE. Congratulations! Your manuscript is now with our production department. 

Kind regards, 

on behalf of

Dr. Petros Isaakidis 

Academic Editor

PLOS ONE